# Observation of reflectionless absorption due to spatial Kramers–Kronig profile

Dexin Ye [1], Cheng Cao[1], Tianyi Zhou[1], Jiangtao Huangfu[1], Guoan Zheng[2,3] & Lixin Ran[1]

As a fundamental phenomenon in electromagnetics and optics, material absorption has been extensively investigated for centuries. However, omnidirectional, reflectionless absorption in inhomogeneous media has yet to be observed. Previous research on transformation optics indicated that such absorption cannot easily be implemented without involving gain media. A recent theory on wave propagation, however, implies the feasibility to implement such absorption requiring no gain, provided that the permittivity profile of this medium can satisfy the spatial Kramers–Kronig relations. In this work, we implement such a profile over a broad frequency band based on a novel idea of space–frequency Lorentz dispersion. A wideband, omnidirectionally reflectionless absorption is then experimentally observed in the gigahertz range, and is in good agreement with theoretical analysis and full-wave simulations. The proposed method based on the space–frequency dispersion implies the practicability to construct gain-free omnidirectionally non-reflecting absorbers.

[1] Laboratory of Applied Research on Electromagnetics (ARE), Zhejiang University, Hangzhou 310027, China. [2] Biomedical Engineering, University of Connecticut, Storrs, CT 06269, USA. [3] Electrical and Computer Engineering, University of Connecticut, Storrs, CT 06269, USA. Correspondence and requests for materials should be addressed to D.Y. (email: desy@zju.edu.cn) or to L.R. (email: ranlx@zju.edu.cn)

As a fundamental phenomenon in electromagnetics and optics, material absorption has been an important topic for centuries. In recent years, research efforts have been focused on the design of inhomogeneous materials that absorb electromagnetic waves in all directions without any reflection[1]. One example is the use of transformation optics technique, which can produce non-reflecting anisotropic materials with arbitrary coordinate transformation[2, 3]. By introducing complex coordinates in the transform, one can design omnidirectionally reflectionless materials that absorb electromagnetic waves incident from all angles[4, 5]. In fact, the perfectly matched layers (PMLs), previously defined mathematically in computational electromagnetics[6, 7], can also be considered as absorptive transformation-optics media[8]. However, during any complex coordinate transform, the generation of a gain region is inevitable[9]. Constrained by the intrinsic causality/stability issue, the physical implementation of an absorptive transformation optics medium together with a gain medium satisfying a strict anisotropic dispersion is challenging. Similarly, non-reflecting absorptive media designed by the parity-time symmetric property have the same requirement of gain regions[10–12].

Different from these methods, a recent research has suggested that a class of isotropic, inhomogeneous one-dimensional (1D) susceptibility profile (permittivity $\varepsilon(x)$ or permeability $\mu(x)$ along the $x$ direction) can be used to design a non-reflecting absorptive material that requires no gain[13]. The only criterion is that this complex susceptibility profile is analytical in the upper or lower half of the complex plane, which consequently satisfies the spatial

**Fig. 1** Wave propagation in an inhomogeneous, dispersive medium. The permittivity profile $\varepsilon(\omega, x)$ of this medium obeys the Kramers–Kronig (K–K) relations in both the frequency domain and the space domain, and its real (**a**) and imaginary (**b**) parts are given by Eq. (2) with parameters $\omega_p = 0.4\omega_0$, $\gamma = 0.03\omega_0$, and $q = 0.5\omega_0$. **c–f** Plots of the $x$-dependent complex permittivity at different frequencies ($0.6\omega_0$, $0.8\omega_0$, $\omega_0$, and $1.2\omega_0$, respectively). **g–j** Simulated $z$-polarized electric field radiated by a line source placed at $x = -8\lambda_0$. Note that the same absolute color scale is used for all figures. While no reflection (standing wave) is seen, the waves are eventually dissipated in different resonance regions of the spatial K–K profiles for different frequencies. **k–n** The intensity drops in dB scale along the $x$ axis for $y = 0$

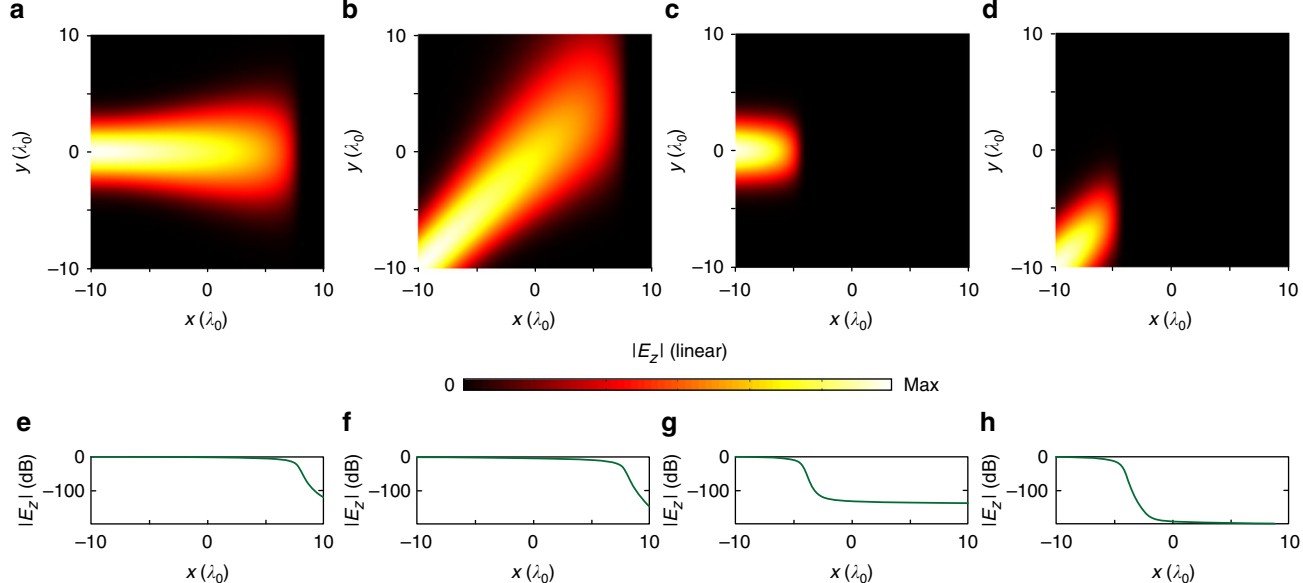

**Fig. 2** Simulations with different locations of Gaussian beam sources. In the simulation, parameters $\omega_p$, $\gamma$, and $q$ in the permittivity profile $\varepsilon(\omega, x)$ described by Eq. (2) are set as $\omega_p = 0.4\omega_0$, $\gamma = 0.03\omega_0$, and $q = 0.5\omega_0$, respectively. Simulations at the frequency of $0.6\omega_0$ (**a**, **b**) and $1.2\omega_0$ (**c**, **d**) are performed, respectively. In **a** and **c**, the Gaussian beam source is located at $(-10\lambda_0, 0)$, and the incident angle of the Gaussian beam with respect to the $x$ axis is 0º. In **b** and **d**, the Gaussian beam source is located at $(-10\lambda_0, -10\lambda_0)$, and the incident angle is 45º. **e-h** The intensity drops along the propagation directions in dB scale are provided, respectively. In all figures, the Gaussian beams are continuously dissipated and eventually absorbed in the resonance region without any observable reflection

Kramers–Kronig (K–K) relations. The design of such profiles initiates a new research direction for non-reflecting wave absorption[14–18]. Without the necessity of using gain media, all functions of PMLs can be implemented in a physical setting.

Based on the space–frequency Lorentzian dispersion proposed in this work, we experimentally implement a permittivity profile that obeys the spatial K–K relations over a broad band by constructing a passive gradient mesoscopic structure. We further demonstrate a wideband, omnidirectionally non-reflecting absorption that is in good agreement with theoretical analysis and full-wave simulation. Based on the concept of dispersion engineering[19, 20], our approach is expected to realize gain-free omnidirectionally reflectionless absorbers.

## Results

**Theoretical analysis.** We begin with the classical Lorentz model describing the dispersion of naturally occurring atomic materials. For a relative permittivity with only one resonance in the positive frequency region, it takes the form of

$$\varepsilon(\omega) = \varepsilon_\infty - \frac{\omega_p^2}{\omega^2 + i\gamma\omega - \omega_0^2}, \tag{1}$$

where $\omega_0$, $\omega_p$, and $\gamma$ are the resonance, plasma and damping frequencies, respectively, $\varepsilon_\infty$ is the static permittivity at infinite frequency, which is generally unity[21]. Such a complex dispersive permittivity is analytical in the upper half of the complex frequency plane, and thus satisfies the K–K relations in the frequency domain[22].

Next, we propose a spatial $x$-coordinate dependence into Eq. (1) by

$$\varepsilon(\omega, x) = \varepsilon_\infty - \frac{\omega_p^2}{\omega^2 + i\gamma\omega - (\omega_0 - qx)^2}, \tag{2}$$

Compared with the fixed resonance frequency $\omega_0$ in Eq. (1), $\omega_0 - qx$ can be considered as an $x$-dependent resonance frequency,

where the constant $q$ determines the change rate of $\omega_0 - qx$ with respect to $x$. Obviously, at any location $x$, this profile satisfies the K–K relations in the frequency domain. In the meantime, at any given frequency $\omega$, it also satisfies the K–K relations in the space domain. In the following, this characteristic will be used in implementing a spatial K–K profile with resonance elements satisfying the proposed space–frequency K–K relations. Note that the spatial K–K profile implemented according to Eq. (2) is broadband by nature.

To demonstrate this relationship between the space- and frequency-domain K–K relations, we show a space–frequency K–K profile in Fig. 1. With parameters $\varepsilon_\infty = 1$, $\omega_p = 0.4\omega_0$, $\gamma = 0.03\omega_0$, and $q = 0.5\omega_0$, the real and imaginary parts of permittivity profile $\varepsilon(\omega, x)$ are shown in Fig. 1a and b, where $\lambda_0$ is the wavelength in free space for frequency $\omega_0$. We see that for a given frequency $\omega$, $\varepsilon(\omega, x)$ exhibits an $x$-dependent spatial Lorentzian dispersion. With increasing frequency, the corresponding resonance region linearly moves to the left. For illustration, Fig. 1c–f shows the $x$-dependent permittivity profiles calculated by Eq. (2) at $0.6\omega_0$, $0.8\omega_0$, $\omega_0$, and $1.2\omega_0$, respectively. It indicates that the strongest absorption would occur at places with different $x$ values for different frequencies.

As discussed in ref. [13], a medium with such a permittivity profile is always reflectionless when probed from one side[13]. To validate this, we perform a full-wave simulation using COMSOL Multiphysics by placing a line source inside the inhomogeneous medium. For the permittivity profiles shown in Fig. 1c–f, the simulated propagation of electric fields is shown in Fig. 1g–j, respectively, where all the line sources radiating $z$-polarized electric fields are placed at the location $x = -8\lambda_0$. To facilitate the analysis and comparison, we also show in Fig. 1k–n the intensity drop in dB scale along the $x$ direction at $y = 0$ simulated for the K–K profile medium (*solid line*) and free space (*dashed line*), respectively. We see that the resonance region acts as a matched absorbing boundary that dissipates the energy of omnidirectional incidences, and thus no visible standing wave effect can be observed on the left side of this region. Compared with the

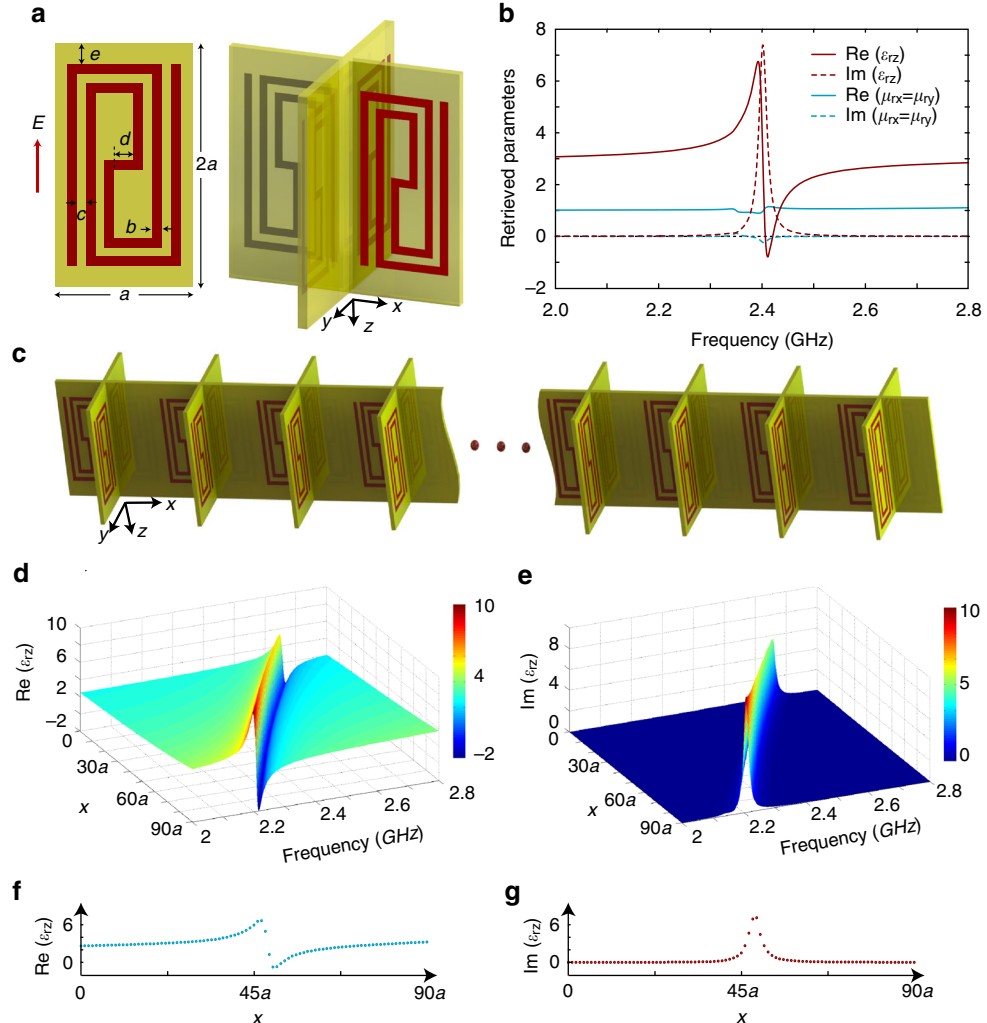

**Fig. 3** Design of the inhomogeneous media with a one-dimensional spatial Kramers–Kronig profile. **a** Geometry of the printed rolled-up wires and the formed two-dimensional (2D) artificial medium. These rolled-up metallic wires (0.018-mm-thick copper with a conductance of $5.986 \times 10^7$ S/m) are printed on a 1-mm-thick dielectric substrate (F4 with a dielectric constant of 2.55 and loss tangent of 0.003). The dimensions $a$ and $2a$ denote the side lengths of the unit cell, $b$ denotes the width of metallic wires, $c$ denotes the distance between metallic wires, $d$ denotes the length of the metallic segment in the center area, and $e$ denotes the distance between the metallic pattern and the upper and lower edges of the unit cell. **b** The retrieved effective constitutive parameters with $a = 6$ mm, $b = c = 0.4$ mm, $d = 0.35$ mm, and $e = 1.11$ mm. The permittivity exhibits the Lorentz resonance model around 2.4 GHz and the permeability is approaching unity. **c** The strip-shaped structure composed of 91 unit cells along the x-axis with different geometric parameter $e$, which changes linearly from 1.6 mm to 0.7 mm with a step of 0.01 mm. **d**, **e** The retrieved real part and imaginary part of permittivity with respect to frequencies and spatial coordinate x, respectively. **f**, **g** The x-dependent real and imaginary parts of permittivity at 2.4 GHz, which satisfies the spatial Kramers–Kronig relations

transmission decay in free space, the absorption due to the resonance region is larger than 120 dB. These results imply a broadband, reflectionless propagation for all angles of incidence. For different frequencies, the propagating waves are finally dissipated in different resonance regions of the spatial K–K profiles, which is in accordance with our theoretical prediction.

Figure 2 further shows the propagation and absorption of Gaussian beams at two different frequencies when the sources are placed at different positions. We see that in all cases, the Gaussian beams are continuously dissipated and finally absorbed in corresponding resonance regions without any observable reflection, and all the intensity drops in such resonance regions are larger than 100 dB.

**Design.** According to the above discussion, the spatial K–K profile can be implemented with different resonance elements

satisfying the frequency-domain K–K relations. Such resonance elements should be at deep-subwavelength scale to obtain a quasi-continuous space–frequency Lorentzian dispersion described by Eq. (2) along the x direction.

In this work, we use a rolled-up metallic wire (0.018-mm-thick copper with a conductance of $5.986 \times 10^7$ S/m) to obtain the desired sub-wavelength electric resonance, as shown in the left of Fig. 3a. These wires are printed on a 1-mm-thick dielectric substrate (F4 with a dielectric constant of 2.55 and a loss tangent of 0.003). Under a z-polarized incidence, an electric resonance would occur when the electric length of the rolled-up wires is around half wavelength. Four such unit cells are rotated to the x, −y, −x, and y directions, respectively, to form an element rotationally symmetric in the y–z and x–z planes, as shown in the right of Fig. 3a.

Next, we perform full-wave simulations to retrieve the constitutive parameters of the artificial medium consisting of

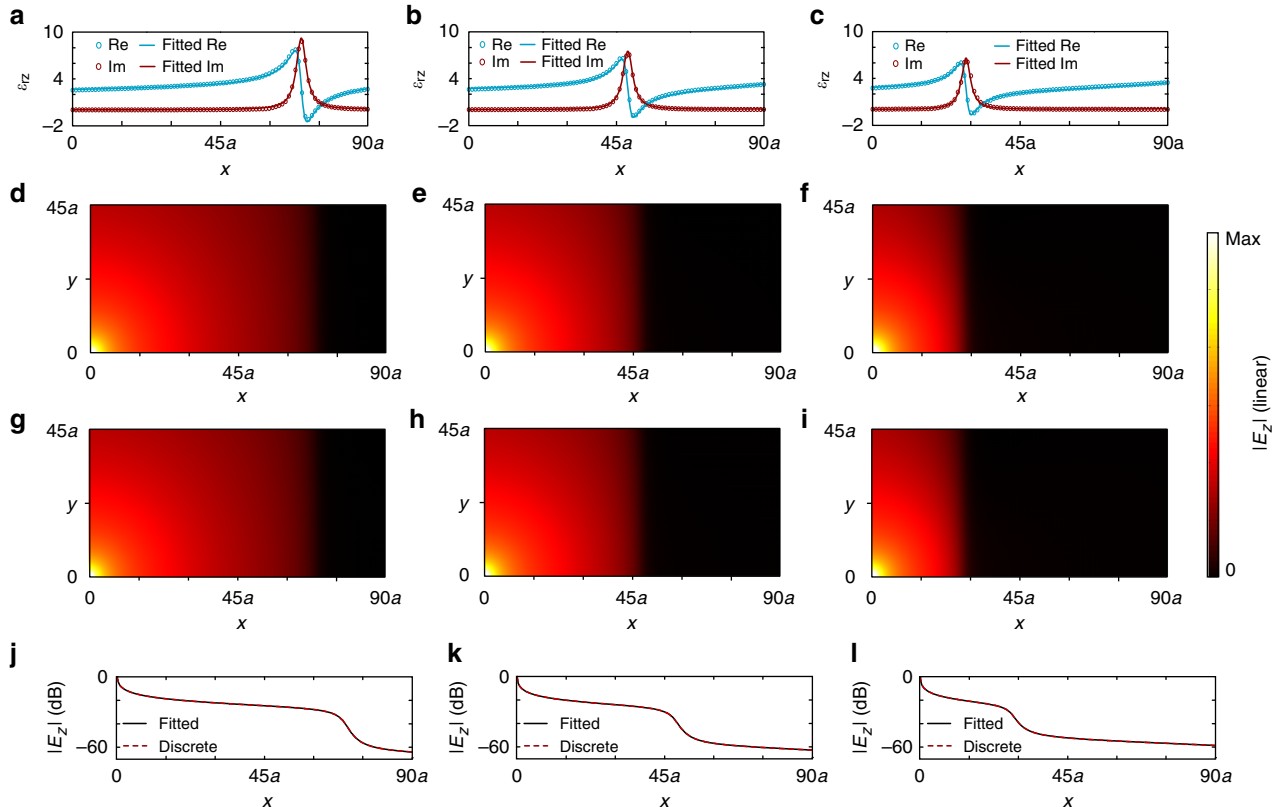

**Fig. 4** Full-wave simulations to the designed one-dimensional inhomogeneous plate. The simulations are performed at 2.3, 2.4, and 2.5 GHz. **a–c** The discrete x-dependent Kramers–Kronig (K–K) profiles (*open circles*) and the corresponding fitted continuous ones (*solid lines*) at 2.3, 2.4, and 2.5 GHz, respectively. **d–f** The simulated amplitudes of z-polarized electric fields when a line source is placed at the origin of the plate with the discrete K–K profiles. **g–i** The simulated amplitudes with the fitted continuous K–K profiles. **j–l** The intensity drops in dB scale along the x axis at y = 0 for the discrete (*open circles*) and fitted (*solid lines*) K–K permittivity profiles. Note that the same absolute color scale is used for all figures from **d–i**

such elements based on the simulation setup and the retrieval algorithm used in reference[23]. The simulation is based on a commercial Maxwell-equations solver, CST Microwave Studio. In the simulation, one unit cell is placed in a parallel-plate waveguide, and the periodic boundaries are used to extend the single unit cell into an infinite slab. The geometric dimensions related to the rolled-up wires were optimized to $a = 6$ mm, $b = c = 0.4$ mm, $d = 0.35$ mm, and $e = 1.1$ mm, so that the electric resonance locates in the design frequency band of 2–3 GHz. The size of this cell is about 1/20 wavelength, which is sufficiently small for constructing a quasi-continuous spatial K–K profile, as shown in the following.

For periodic structures consisting of such unit cells, the effective permittivity and permeability retrieved from the simulated scattering data are plotted in Fig. 3b. As expected, a Lorentz resonance region is observed at 2.4 GHz for the retrieved permittivity. In the meantime, a weak "parasitic" resonance for the retrieved permeability is observed in the same region due to the coupling of the electric and magnetic fields in the same unit cell. It follows an inverse Lorentz resonance model[24, 25]. Therefore, both the effective permittivity and permeability satisfy K–K relations in the frequency domain.

In our design, the dimension $e$ can be tuned to change the location of the resonance region. As such, a series of resonance elements can be achieved at different resonance regions. We line up 91 such elements along the x-axis from $x = 0$ to $x = 90a$ with a uniform periodicity of $a = 6$ mm. From the 1st to the 91st cells, the dimension $e$ of each element linearly decreases from 1.6 mm to 0.7 mm, so that the resonance frequency of each element also linearly decreases along the x direction. Such 91 elements are then

duplicated along the y direction with the same periodicity of $a$, constructing a two-dimensional (2D) plate. In this case, the obtained space–frequency Lorentzian dispersion can be considered as:

$$\varepsilon(\omega, x) = 3 - \frac{(0.57 \times 10^9)^2}{\omega^2 + i\omega \times (1.8 \times 10^7) - [2.657 \times 10^9 - (0.5 \times 10^7/a) \times x]^2}, \quad (3)$$

where $x$ and $\omega$ range from 0 to 90a and 2 to 2.8 GHz, respectively. Figure 3d and e shows the real and imaginary parts of this permittivity profile with respect to the frequency $\omega$ and location $x$, respectively. We see that the permittivity of this inhomogeneous plate exhibits 1D K–K profiles in both the space and frequency domain, in good agreement with the analytical results shown in Fig. 1. Note that the 1D spatial profile of the plate consisting of discrete elements is also discrete. For 2.4-GHz frequency, the discrete profiles are shown in Fig. 3f and g. Since the periodicity of the elements is around $\lambda_0/20$, the macroscopic spatial dispersion would be very close to a continuous one, as shown in the following. It should be noted that the K–K profile medium consisting of such elements would only work for electric fields polarized along the z direction.

**Full-wave simulation.** In order to observe the wave propagation in the designed plate, we performed a full-wave simulation whose results are presented in Fig. 4. In the simulation model, a rectangular plate consisting of 46 identical 91-element strips in the y direction is truncated with PML, and a line source is placed in the bottom left corner.

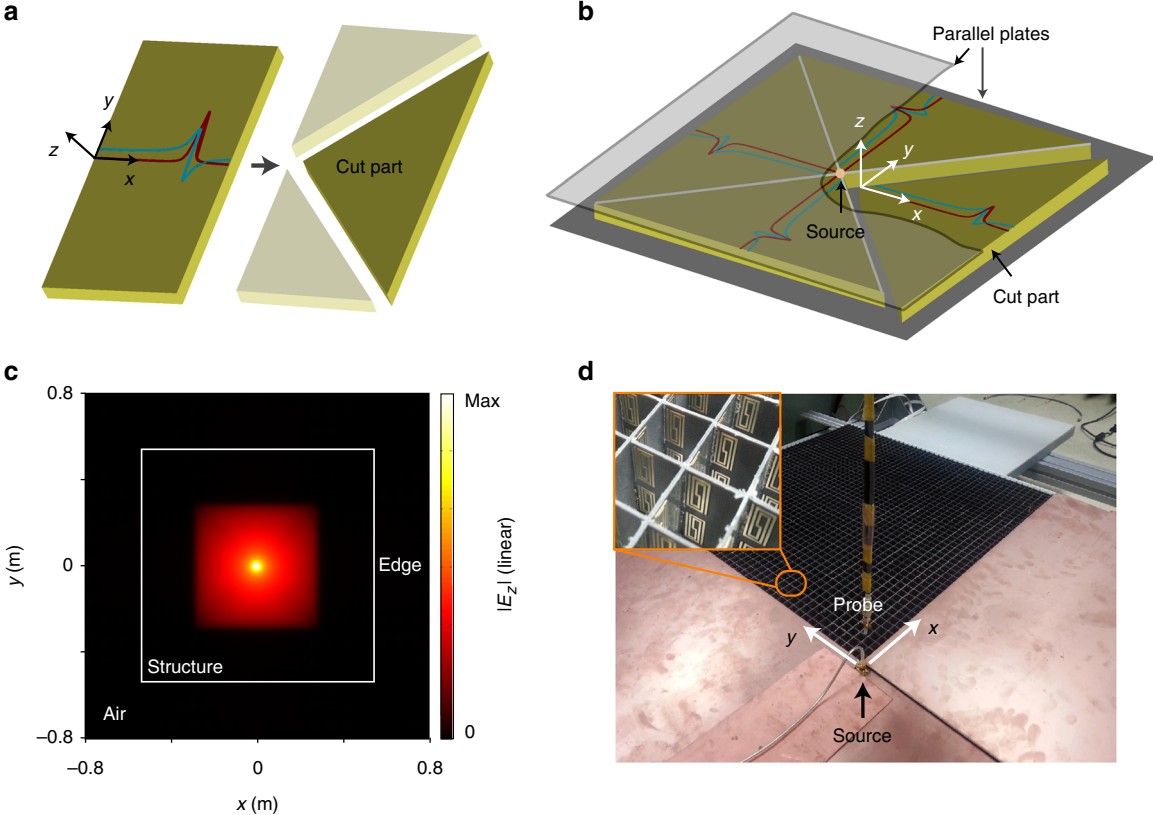

**Fig. 5** Fabrication of the one-dimensional inhomogeneous sample and experimental setup. **a** Cutting off an isosceles right-triangle part from the original rectangular plate consisting of 2 × 91 × 182 unit cells. The real (*blue line*) and imaginary (*red line*) parts of the inhomogeneous permittivity of the cut part satisfy the designed Kramers–Kronig (K–K) profile along the x direction. **b** Reassembling of four identical cut plates into a larger symmetric square plate. Seen from the origin, the real (*blue lines*) and imaginary (*red lines*) parts of the inhomogeneous permittivity satisfy the same K–K profile along the x, y, –x, and –y directions, respectively. This plate is then sandwiched by two parallel copper-plated boards so that only a transverse electromagnetic mode can exist in the plate. Both boards are slightly larger than the plate. **c** Full-wave simulation of the sample shown in **b** at 2.4 GHz when the radiative source is placed in the center of the square plate. In the model, a free space boundary was used at the cutting edge, and a perfectly matched layer (PML) boundary was used to enclose the entire simulation region. **d** Photograph of the actual experimental setup. A small monopole probe with omnidirectional radiation is placed at the center of the square plate. Another identical monopole is sequentially placed in each cell to measure the amplitude and phase of the electric field

First, the discrete spatial K–K profiles retrieved at the 2.3-, 2.4-, and 2.5-GHz frequencies are shown in Fig. 4a–c (*dotted lines*), respectively. For comparison, the fitted continuous profiles are also shown in Fig. 4a–c (*solid lines*). To examine the effect due to the discrete K–K profile, two simulations were performed for discrete and continuous profiles, respectively.

Figure 4d–f and 4g–i show the wave propagation in the designed plate with the discrete and continuous profiles, respectively. Figure 4j–l shows the intensity drops in dB scale along the x direction at y = 0 for both the discrete and fitted K–K permittivity profiles. We see that with the 1/20 $\lambda_0$ spatial sampling, the simulated results based on the discrete profile are almost the same as those obtained based on the continuous ones. This implies that the used elements have been sufficiently small to construct the desired spatial profile. In the meantime, no standing wave effect can be observed in both cases, implying a reflectionless propagation in a wide frequency band of 2.3–2.5 GHz. For different frequencies, the propagating waves are finally dissipated in different resonance regions, agreeing with the analytical results in Fig. 1.

**Measurements**. To experimentally verify the simulated results, we fabricated four identical rectangular plates, each consisting of 2 × 91 × 182 elements. The 91 elements along the x direction were

designed to have the same gradient profile as that in the simulation. These elements were replicated along the y and z direction so that each plate has the same x-dependent K–K profile. In order to observe the reflectionless propagations in an experimental setup similar to the simulated one in Fig. 4, an isosceles right-triangle part was cut from each plate, as shown in Fig. 5a. The cut parts were reassembled into a larger 2D symmetric square plate in Fig. 5b, in which two unit cells are stacked in the z direction. This plate was sandwiched by two parallel copper-plated boards so that only a transverse electromagnetic wave mode can exist in the plate[26]. Both boards are slightly (around one wavelength) larger than the plate. By doing so, the radiated electromagnetic waves would see the same K–K profiles in both ±x and ±y directions if the radiative source is placed in the center of the square plate.

Figure 5c shows the simulated propagation of the electric field radiated by a line source placed at the center of the square plate. For the 2.4 GHz radiation, no reflections can be observed in all directions even when the edges of this sample are mismatched to the air. The propagating waves are almost completely dissipated in the resonance region before they can arrive at the edges.

In the actual measurement, a small monopole antenna with an omnidirectional radiation was placed at the center of the square plate[27]. Another identical monopole was sequentially placed in each cell to measure the amplitude and phase of the electric field.

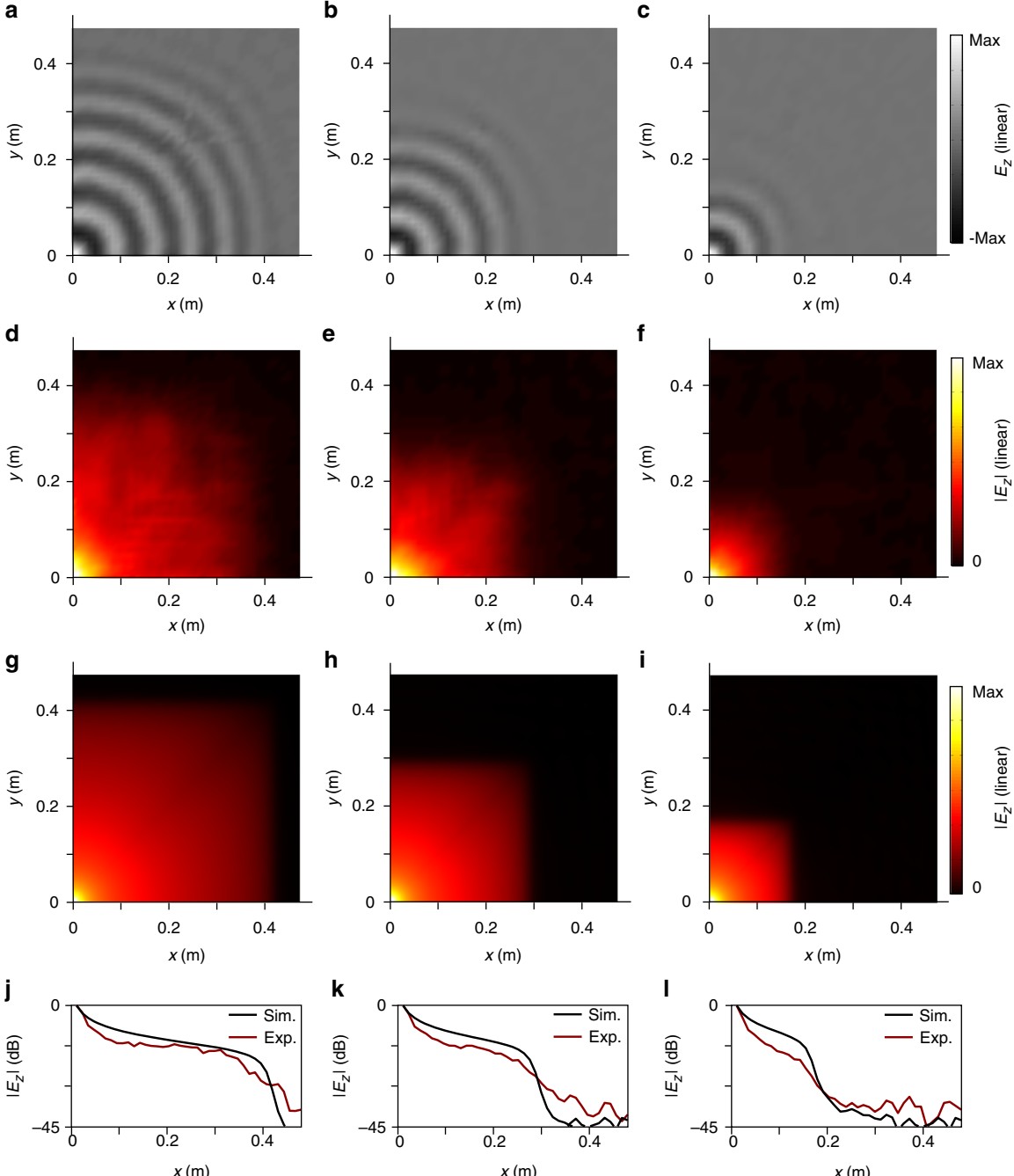

**Fig. 6** Experimental results and analysis. **a–c** Instantaneous snapshots of the electric fields measured at 2.3, 2.4, and 2.5 GHz, respectively. All the measured data are normalized by the amplitude and phase of the electric field measured at one unit cell nearest to the source monopole. **d–f** Amplitudes of the electric fields measured at the same frequencies. For comparison, **g–i** show the corresponding simulations under the same conditions. **j–l** Comparison between the measured and simulated intensity drops in dB scale along the $x$ axis at $y = 0$

To minimize the potential influence, when one cell was measured, the other cells were all covered by copper-plated boards. For each cell, an Agilent E8361A vector network analyzer was used to measure the transmission parameter. In order to calibrate the response of the monopoles, all the measured data were normalized by the amplitude and phase of the electric filed measured at one unit cell nearest to the source monopole. A photograph of the actual experimental setup is shown in Fig. 5d.

The propagation of electric fields in a quadrant of the square plate measured at 2.3, 2.4, and 2.5 GHz is shown in Fig. 6. Figure 6a–c shows the measured instantaneous electric fields. We see that the propagating wave is continuously dissipated along the radiation directions without any observable standing wave effect. Figure 6d–f and 6g–i show the amplitude distributions of the measured and simulated electric fields, respectively. We see that for different frequencies, the propagating waves are eventually dissipated at different locations. The measured locations agree with simulated ones.

For comparison, Fig. 6j–l shows the measured and simulated intensity drops at the same frequencies. We see that in the three cases, the experimental intensity drops measured at the corresponding edge are 40 dB, 41 dB, and 39 dB for 2.3, 2.4, and 2.5 GHz, respectively, while all the simulated intensity drops are around 45 dB.

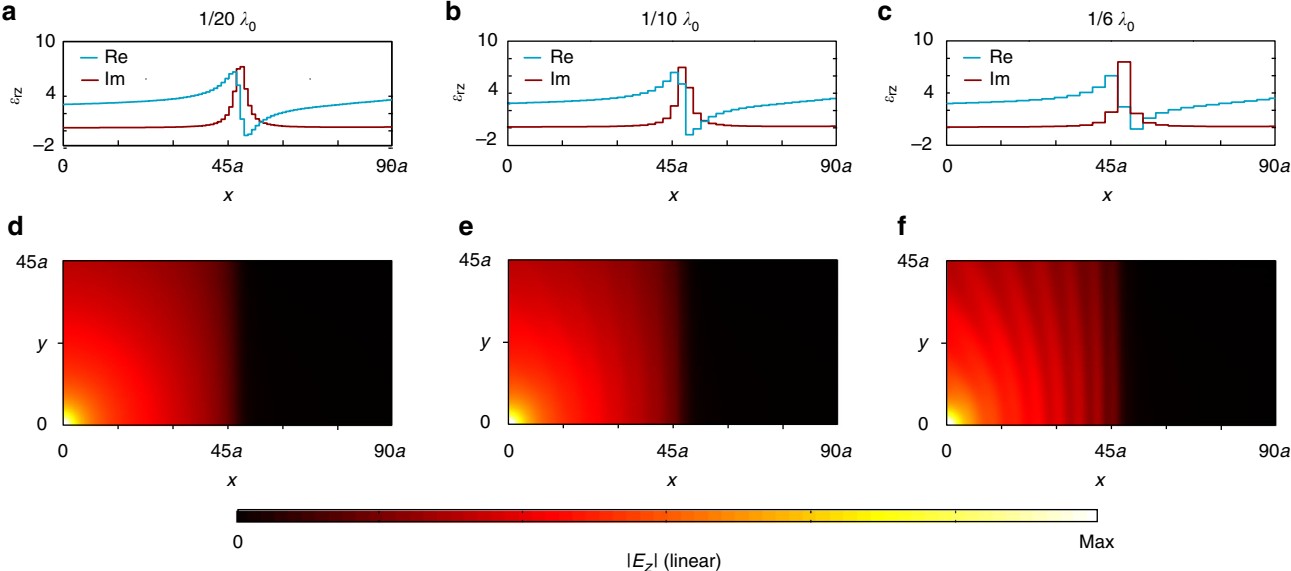

**Fig. 7** Performances of discrete spatial Kramers–Kronig profiles with different discretization steps analyzed at 2.4 GHz. **a–c** The discrete *x*-dependent Kramers–Kronig profile with discretization steps of $\lambda_0/20$, $\lambda_0/10$, and $\lambda_0/6$, respectively. **d–f** The corresponding simulated amplitudes of the electric fields. The units of the *x* and *y* axis are the width *a* of the unit cell, as shown in Fig. 3a. Note that the same absolute color scale is used in these figures

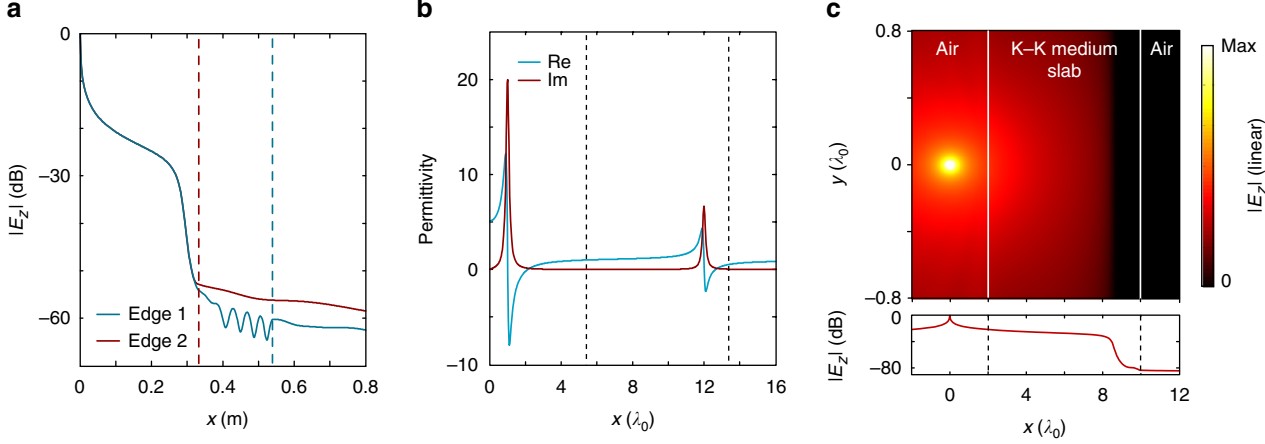

**Fig. 8** Impact of finite-size truncations. **a** Standing wave effect simulated at 2.4 GHz when the designed Kramers–Kronig (K–K) profile medium is cut off at different locations on the right side of the resonance region. The *dashed red line* indicates the location of edge 1 located at *x* = 0.54 m, where the permittivity is 3.3. The *dashed blue line* indicates the location of edge 2 at *x* = 0.33 m, where the permittivity is 1. The *solid blue* and *red* curves show the simulated standing wave effects due to edges 1 and 2, respectively. **b** Construction of a K–K profile medium slab for incidences from outside. The *left dashed line* indicates the front edge of the slab located at *x* = 5.45$\lambda_0$, where $\varepsilon_r$ = 1. The *right dashed line* indicates the back edge of the slab located at *x* = 13.45$\lambda_0$, where $\varepsilon_r$ = 0.55. The thickness of the designed slab is 8$\lambda_0$. **c** Simulated results for the K–K profile medium slab described in **b** when a point source is placed 2$\lambda_0$ away from the front edge

The experimental results also show that small defects in the fabrication and assembly of the unit cells could potentially induce a local field disturbance. Examples can be seen in Fig. 6d–f, where electric field disturbance is observable in the region near the seams between the adjacent cutting plates. However, this kind of disturbance due to unit cell defects does not impact the understanding of the experimental results.

## Discussion

The above measurements verified that the spatial K–K profile medium, which is by far the only inhomogeneous medium that can exhibit omnidirectional, reflectionless absorption without gain, can be physically implemented. Based on the concept of space–frequency Lorentzian dispersion proposed in this work,

wideband absorption can also be practically obtained. A unique characteristic of this K–K profile medium is that it has an omnidirectionally matched absorbing boundary.

In the physical implementation, it is important to ensure that the impact to the continuity of the spatial K–K profile due to the discretization of resonance cells should be minimized[28]. As shown in Fig. 4, the reflectionless absorption simulated with $\lambda_0/20$-sized resonance cells has not an observable difference with that simulated with continuous K–K profiles. However, as shown in Fig. 7, when the size of the cells is reduced to $\lambda_0/10$ and $\lambda_0/6$, weak standing wave effects become observable due to the increased discretization. The maximum standing wave ratios calculated from Fig. 7d–f are 1.02, 1.16, and 1.85, respectively. This further confirms the importance of using deep subwavelength cells in the implementation of spatial K–K profiles.

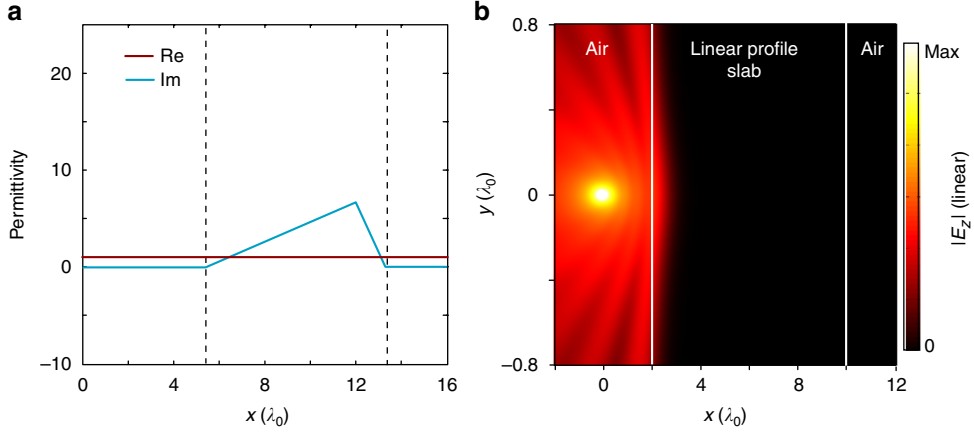

**Fig. 9** Reflection of a linear profile slab. **a** The linear permittivity profile slab with the same thickness and variation range of the imaginary part of permittivity as those shown in Fig. 8b. The real part of the linear permittivity profile is set as a constant value of 1, so that it can perfectly match the real part of the permittivity of air. The imaginary part is linearly increased from 0 (at the front edge located at $x = 5.45\lambda_0$) to the maximum value of 6.7 at $x = 12\lambda_0$. Here both the maximum value and its location are the same as those in Fig. 8b. It is then linearly decreased from 6.7 to 0 at the back edge of the slab located at $x = 13.45\lambda_0$. This configuration is the best case for an absorbing slab with a linear permittivity profile to obtain an optimal reflection performance. **b** The simulated wave reflection due to the linear profile slab

In practical applications, there are two different usages of finite-sized K–K profile media. The first scenario is to absorb waves radiated from inside. In this case, the absorber can be designed following the method used in our experimental setup. As discussed, when the permittivity profile is cut on the right-hand side of the resonance region, for radiations coming from its left side, this region can be considered as an omnidirectionally matched absorbing boundary. Multiple such boundaries can be used to enclose the space containing radiative sources.

Figure 8a shows the full-wave simulation of the standing wave effects due to different cutting edges. The edge 1 is located $x = 0.54$ m, which is the case in our experimental setup shown in Fig. 5c. The corresponding relative permittivity $\varepsilon_r$ is around 3.3. Edge 2 is located at $x = 0.33$ m, where $\varepsilon_r$ is around 1. The *blue* and *red curves* show the simulated standing wave effects due to edges 1 and 2, respectively. We can see that while there is no visible standing wave near edge 2, the mismatched permittivity at edge 1 does produce notable standing waves. In both cases, there is no visible impact to the electric field on the left side of the resonance region.

The second scenario is to absorb waves incident from outside. In this case, in order to obtain a slab-like absorber, the permittivity profile should be cut off on both sides of the resonance region. Particularly, it should be cut on its left side at the place where the relative permittivity is close to 1.

We show in Fig. 8b how to obtain such a finite-size K–K medium slab based on the concept of dispersion engineering[19]. By constructing a K–K permittivity profile containing two resonance regions, i.e., $\varepsilon(x) = 1 - x_{p1}^2/(x^2 + i\gamma_1 x - x_1^2) - x_{p2}^2/(x^2 + i\gamma_2 x - x_2^2)$, a unity relative permittivity must exist between locations $x_1$ and $x_2$. If $x_{p1} = 2\lambda_0$, $\gamma_1 = 0.2\lambda_0$, $x_1 = 1\lambda_0$, $x_{p2} = 4\lambda_0$, $\gamma_2 = 0.2\lambda_0$, and $x_2 = 12\lambda_0$, the location with $\varepsilon = 1$ is at $x = 5.45\lambda_0$. If the front edge is cut off at this location, and the back edge can be cut off at a location on the right side of the second resonance region, such as at $x = 13.45\lambda_0$ where $\varepsilon_r = 0.55$, a slab with a thickness of $8\lambda_0$ can be obtained.

Figure 8c shows the full-wave simulation to this slab when a point source is placed $2\lambda_0$ away from the left edge. We see that there is no visible standing wave effect due to the front cutting edge.

In practice, a linear gradient index has been widely used in microwave engineering. Examples include the impedance matching obtained by continuous impedance transform[29] and the absorbing pyramids used in anechoic chambers. Compared with the case of Lorentzian dispersion, the most significant difference is that the linear gradient index does not have a resonance region. As such, the linear gradient index medium does not have a matched absorbing boundary that can exhibit reflectionless absorption over a short distance. This is the reason why a practical continuous impedance transform segment always requires a sufficient length so as to decrease the slope of the linear gradient index.

Figure 9a shows the linear profile slab with the same thickness and the same variation range of the imaginary part of permittivity as those shown in Fig. 8b. Note that in Fig. 9a, the real part has been set to 1, which presents the best case for an absorbing medium with linear profile index. Figure 9b shows the reflection due to such a linear profile slab. We can see that a significant reflection can be observed. Further simulation shows that only when the thickness increases to ~90 wavelengths, the reflection can be reduced to a negligible level.

In summary, we have experimentally verified the omnidirectional reflectionless absorption due to the spatial K–K profile of constitutive parameters. The proposed method based on the space–frequency Lorentz dispersions can be used to construct broadband inhomogeneous materials with desired spatial K–K profiles. Although we demonstrated our experiment at microwave frequencies, our method can also be used at higher frequencies up to the optical regime. Our work implies the practicability to artificially fabricate broadband, omnidirectionally non-reflecting absorbers without gain regions.

## Methods

**Full-wave simulations.** In this work, two commercial software packages, COMSOL Multiphysics and CST Microwave Studio, are used in different full-wave simulations.

In the theoretical analysis, simulations for all the electric field propagation are performed using COMSOL Multiphysics, taking the unique advantage of its high-efficient computation for 2D full-wave simulations. The analysis type is set as "Harmonic propagation", and the "Stationary solver" is chosen. In all simulations, the maximum number of iteration is set as 25, the size of the mesh is set as $\lambda_0/50$, and all the simulated domains are truncated by PML boundaries.

In the material design, the scattering parameters of the three-dimensional unit cells are simulated using CST Microwave studio. In all the simulations, the "FSS-Unit Cell template" in the "Frequency domain solver" is chosen. The accuracy of the "tetrahedral mesh" is set as 0.0001%, the simulation steps per wavelength are

set as 10 in the "Mesh density control", and the minimum number of steps is set as 20.

**Data availability**. The data that support the findings of this study are available from the corresponding authors on request.

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

## Acknowledgements

This work is supported by the NSFC under grants 61401393 and 61131002, the China CASC foundation, the Fundamental Research Funds for the Central Universities under grant 2017QNA5006, and the Sichuan Key Research Project.

## Author contribution

D.Y. and L.R.: Designed the research; D.Y., C.C., T.Z., J.H., and G.Z.: Performed the research. All authors contributed to data interpretation and the composition of the manuscript.

## Additional information

**Competing interests:** The authors declare no competing financial interests.

