## [Peer Review File · Nature Communications]

Reviewers' comments:

Reviewer #1 (Remarks to the Author):

This paper demonstrates a practical implementation of a K-K non-reflective surface. The paper is very well-written, and the graphs are clear of understanding. In terms of quality of the text, the paper fulfill the standards of Nature publications.

From the scientific point of view, I found the paper sounded. The fact that K-K relations are followed at different frequencies following the Lorentz model is very interesting. This makes the structure broadband, and it opens a lot of applications for the structure.

Critical points:

- In the introduction, the authors started speaking about transformation optics. I missed the point of this paper and its relation to transformation optics. I think this is misled.

-All the electric fields are plotted from 0 to max. However, we do not know if the scale is linear or in dB. This is very important because the level of the fields could possibly be very low. Potentially, results plotted in linear scale could significantly appear to our eyes better than in reality.

- I would recommend to the authors one simulation more in one of the graphs. They claimed that this absorber is broadband, which is demonstrated. But they claimed that is 'omnidirectional' in the abstract. However, this is not further studied in the paper. Maybe a simulation with sources at different locations could help to support their claim and to improve the quality of their paper. I understand that the fact that the source produces cylindrical waves (it is a point source) helps to this demonstration, but I think that further simulations could help to the authors to make clear their point.

- I would encourage to the authors to answer the following questions: How many reflections could you get with a conventional absorber material? How this technique can help to produce better absorbers? Maybe this question sounds as it is at a very high level, but I believe that readers of Nature Communications would like to know the authors' opinion about the potential of the technique in comparison with existing and easy implementations (broadband and omnidirectional), as conventional absorbers. These absorbers could be also multilayer: 2 or 3 different values of real epsilon with high imaginary part.

Reviewer #2 (Remarks to the Author):

In the manuscript 'Observation of Reflectionless Absorption Due to Spatial Kramers-Kronig Profile' the absorption of a line absorber embedded in an absorbing material with a resonant permittivity profile satisfying the spatial Kramers-Kronig relations is investigated experimentally (with corresponding numerical simulations).

It is based on the theoretical and numerical work published in [9].

The manuscript is generally well written, but the authors could not make evident the significant impact to justify publication in a high-rank journal like Nature Communications. For this the authors should make clear why such an absorber is mature compared to the numerous absorbers already used in the microwave community. At this stage, I feel that I cannot give a positive recommendation to accept this article for publication. If the authors can convincingly show that this new type of absorbers might be better than other already existing absorbers even only for a few specific applications. Therefore addressing question of bandwidth versus necessary thickness of absorbers, minimal slope of spatial vs frequency position related to the number of cells /wavelength etc are necessary.

I'm missing the discussion on finite size effects as they were done in the supplementary of [9]. Experimentally a cut of the permittivity is at the end of the material. This is leading to reflections and a standing wave part. Can this be given in numbers i.e. dB or % of energy. If the wave would come from outside of the absorbing metamaterial, like in an anechoic chamber, what would be the reflection coming from this interface?

Specific remarks:

page 1:

- very recent research -> recent research
- in a good agreement -> in good agreement
- mathematically speaking eq. 2 has two resonance: $\omega = +$ and $-\omega_{\lambda_0}$

page 2:

- figure 1: Why is the color scale in (b) stating for the imaginary part at -2 when the lowest values is 0, also it indicates negative values for $\text{im}(\epsilon)$ which is not possible. It would be helpful to plot the cut at $y=0$ for the $|E_z|$ maybe in log scale to indicate how high the transmission is (in dB) an important quantity for an absorber. Especially as in an experiment finite size problems will occur. Also the decay that is due to the point/line source should be indicated ($1/R$) as oppose to no decay for a plane wave. What are the boundary conditions (at $\pm 10\lambda_0$) for the calculations (pml)?

page 3:

- Same problem with color scale in figure 2e as in figure 1b
- Parameters of the obtained Lorentz lines should be given

page 5:

- The structure itself is not rotational invariant but is only invariant on rotations of 90° . Might not be clear to all readers why its rotationally invariant for the field.

page 7:

- How close can the line source be placed to the resonant part?

page 8:

- Are the bottom and top plate larger than the metamaterial? When yes how many λ s to the border? Are other absorber used or is the whole setup in free space?
- How is direct coupling by evanescent wave taken into account from 1 monopole antenna (source) to the scanning monopole antenna if they are close?

page 9:

- The simulations shown in figure 5c use which boundary conditions (pml or free space to be more adjusted to the experiment).

Reviewer #3 (Remarks to the Author):

The manuscript demonstrates experimentally at microwaves the recent theoretical finding (from the authors) that absorption over a range of frequencies can happen without reflection in a passive inhomogeneous media provided the permittivity profile of the medium satisfies the spatial Kramers-Kronig (K-K) relations. The topic may be interesting not only from the physics (fundamental) point of view, but also from an engineering (application) perspective. The manuscript is pedagogically written and the work is sound. The measurements support nicely the theoretical frame. It is for these reasons that I am inclined to recommend its publication in Nature Communications. However, the contribution to the state of the art of this work is simply the

experimental demonstration; theory and simulations are not new; I wonder if this is enough for Nature Communications.

Next I list in order of importance the points that the authors should address in the revision version to have my full support.

The authors suggest that this work should have an impact in computational electromagnetics, namely for PML. However, the work does not discuss the common scenario where PML is used: an interface between the PML and a region with fields. As a matter of fact, if the excitation is outside the inhomogeneous medium, all the properties are lost. I think this should be discussed to avoid misleading the readers.

Can the authors please quantify the drop in intensity for each frequency? For example, what is the intensity at the edges (Fig. 5) compared to the excitation point? dB drop of?

The authors should discuss the potential coupling between consecutive elements that may modify the results shown in Fig. 2d,e. Can the authors unambiguously state that local periodicity condition applies in their design?

The manuscript does not give enough technical details (material properties, boundary conditions, mesh, etc.) about the simulations to replicate the results precisely. Please, add such details (perhaps as a supplementary material). How many unit cells are stacked in z ? The text does not say anything and one may think it is just one, but one can see at least two in Fig. 4d. Is the response of the monopole and probe de-embedded in the results? What is the experimental calibration procedure?

The authors claim that only TEM modes are supported. How do they rule out modes like magnetoinductive waves (this is related to the potential mutual coupling between consecutive elements)?

The design is for Ez-polarized waves and it will not work for Hz-polarized waves. This should be stated clearly in the text.

Can the authors quantify the standing wave ratio (SWR) for each case in Fig. 6?

Please, add the dimensions units in 4c and 5.

Reviewer #1

This paper demonstrates a practical implementation of a K-K non-reflective surface. The paper is very well-written, and the graphs are clear of understanding. In terms of quality of the text, the paper fulfills the standards of Nature publications.

From the scientific point of view, I found the paper sounded. The fact that K-K relations are followed at different frequencies following the Lorentz model is very interesting. This makes the structure broadband, and it opens a lot of applications for the structure.

Thank you for the encouraging comments.

Critical points:

In the introduction, the authors started speaking about transformation optics. I missed the point of this paper and its relation to transformation optics. I think this is misled.

Before the report of the K-K profile medium, transformation-optics technique has been used to generate media that exhibit omnidirectionally reflectionless absorption [R1]. In fact, the well-known perfectly matched layer (PML) can be considered as a transformation-optics medium.

Among many inhomogeneous PML models, the simplest one is the uniaxial PML model. It is described by $\epsilon_r = \mu_r = \text{diag}(a(\bar{r}), a(\bar{r}), 1/a(\bar{r}))$, where space dependent $a(\bar{r})$ is complex with a positive imaginary part. Consequently, the imaginary part of $1/a(\bar{r})$ has to be negative, implying gain along the optical axis.

From transformation optics point of view, this PML model is a result of a complex coordinate transform ($x' = x + i\Delta x/\omega$) to free space. During any complex coordinate transform, the generation of a gain factor is inevitable [R2]. Therefore, **absorptive** transformation optics media are essentially gain media. This is also the essential reason why the K-K profile medium is significant. It is by far the only exception that exhibits omnidirectionally reflectionless absorption without gain.

In the revision, we have further clarified this connection. Please refer to our revised manuscript for details.

[R1] Popa, B-I. & Cummer, S. A. Complex coordinates in transformation optics. Phys. Rev. A 84,063837 (2011).

[R2] Odabasi, H., Teixeira, F. L. & Chew, W. C. Impedance-matched absorbers and optical pseudo black holes. J. Opt. Soc. Am. B 28, 1317-1323 (2011).

-All the electric fields are plotted from 0 to max. However, we do not know if the scale is linear or in dB. This is very important because the level of the fields could possibly be very low. Potentially, results plotted in linear scale could significantly appear to our eyes better than in reality.

In our manuscript, all the electric fields were plotted in linear scale.

Fig. R1. Comparison between the figures in linear and dB scales.

We agree with you that dB scale is normally better for showing the intensity change with a wider range. Figure R1 shows the comparison between the cases of linear scale (a) and dB scale (b). Since the intensity drop is as large as 185 dB and the abrupt intensity change mainly occurs in the resonance region, using dB scale would result in the loss of detailed information.

Fig. R2. Updated figure 1c in the revision.

Following your (and the Reviewer 2's) suggestion, we have added 4 new subfigures in the bottom of Fig. 1c to show the abrupt intensity drop in dB scale along the x axis at $y = 0$ (bottom of Fig. R2). We can see that the resonance region acts as a matched boundary that dissipates the energy of omnidirectional incidences.

- I would recommend to the authors one simulation more in one of the graphs. They claimed that this absorber is broadband, which is demonstrated. But they claimed that is 'omnidirectional' in the abstract. However, this is not further studied in the paper. Maybe a simulation with sources at different locations could help to support their claim and to improve the quality of their paper. I understand that the fact that the source produces cylindrical waves (it is a point source) helps to this demonstration, but I think that further simulations could help to the authors to make clear their point.

Thank you for this suggestion.

Fig. R3. Simulations with different locations of Gaussian beam sources at the frequency of $0.6\omega_0$.

Following your suggestion, we have added a new figure (Fig. 2 in the revised manuscript) where Gaussian beam sources at the frequency of $0.6\omega_0$ are placed at different positions for incidences. For your convenience, this figure is shown in Fig. R3. We can see that for both cases, the Gaussian beams are continuously dissipated and finally absorbed in the resonance region without any observable reflection.

- I would encourage to the authors to answer the following questions: How many reflections could you get with a conventional absorber material? How this technique can help to produce better absorbers? Maybe this question sounds as it is at a very high level, but I believe that readers of Nature Communications would like to know the authors' opinion about the potential of the technique in comparison with existing and easy implementations (broadband and omnidirectional), as conventional absorbers. These absorbers could be also multilayer: 2 or 3 different values of real epsilon with high imaginary part.

Thank you for this suggestion.

A very special characteristic of this K-K profile medium is that it has an omnidirectionally matched absorbing region. In theory, all energy of a wave entering into this region from its left side will be absorbed without any reflection. As seen in Fig. R2, a sharp drop of incident energy up to 120-dB can be observed in this narrow region. Based on the concept of “space-frequency Lorentzian dispersion” proposed in this work, wideband absorption can also be practically obtained.

In fact, the experimental setup shown in Fig. 5 of our manuscript illustrates how this K-K profile medium can be used to construct an absorber to dissipate the radiation of an omnidirectional source. In theory, this kind of absorber cannot be implemented by existing absorbing materials.

For conventional inhomogeneous absorbers, such as multi-layered absorbers, reflections must exist for oblique incidences. For conventional homogeneous absorbers, reflections must exist at the boundary due to the mismatch between the lossy medium and the air.

In the revision, we have added these discussions. Please refer to our revised manuscript for details.

Reviewer #2

In the manuscript 'Observation of Reflectionless Absorption Due to Spatial Kramers-Kronig Profile' the absorption of a line absorber embedded in an absorbing material with a resonant permittivity profile satisfying the spatial Kramers-Kronig relations is investigated experimentally (with corresponding numerical simulations). It is based on the theoretical and numerical work published in [9].

The manuscript is generally well written, but the authors could not make evident the significant impact to justify publication in a high-rank journal like Nature Communications. For this the authors should make clear why such an absorber is mature compared to the numerous absorbers already used in the microwave community. At this stage, I feel that I cannot give a positive recommendation to accept this article for publication. If the authors can convincingly show that this new type of absorbers might be better than other already existing absorbers even only for a few specific applications. Therefore addressing question of bandwidth versus necessary thickness of absorbers, minimal slope of spatial vs frequency position related to the number of cells /wavelength etc are necessary.

Thank you for the insightful comments.

Concerning the significance of this work, we proposed the concept of “space-frequency Lorentzian dispersion” that can be used to practically implement a wideband K-K profile medium. Our experimental results also verified the theoretical predictions made in reference 13. As we have replied to Reviewer 1, this K-K profile medium is by far the only inhomogeneous medium that can exhibit omnidirectionally reflectionless absorption without gain.

Compared with existing absorbing materials, a unique characteristic of this medium is that it has an omnidirectionally matched absorbing region. In theory, all energy of a wave entering into in this region from its left side will be absorbed without any reflection, no matter the angle of incidence. As shown in the revised Fig. 1, a sharp drop of incident intensity up to 120-dB can be observed in this narrow region. To the best of our knowledge, this kind of absorption cannot be implemented by existing absorbers.

In the revision, we have further clarified and emphasized these points. Please refer to our revised manuscript for details.

I'm missing the discussion on finite size effects as they were done in the supplementary of [9]. Experimentally a cut of the permittivity is at the end of the material. This is leading to reflections and a standing wave part. Can this be given in numbers i.e. dB or % of energy. If the wave would come from outside of the absorbing metamaterial, like in a anechoic chamber, what would be the reflection coming from this interface?

Thank you for raising this point.

Concerning the usage of the finite-sized K-K profile medium, there are two different scenarios.

The first scenario is to absorb waves radiated from inside. In this case, the absorber can be designed following the method used in our experimental setup. As discussed, when the permittivity profile is cut on the right side of the resonance region, for radiations coming from its left side, this region can be

considered as an omnidirectionally matched absorbing boundary. Multiple such boundaries can be used to enclose the space containing radiative sources.

Fig. R4. Simulated results of the reflection when the designed K-K medium is cut at different locations at 2.4 GHz. The blue line is the result for our experiment setup where the K-K medium is cut at $x = 0.54$, whose permittivity is around 3.3 at the cut edge, and the red line shows the result if we cut the medium at $x = 0.33$ whose permittivity is around 1.

Figure R4 shows the full-wave simulation of the standing wave effects due to different cutting edges. The edge 1 is located $x = 0.54$ m, which is the case in our experimental setup shown in Fig. 5c in the revision. The corresponding relative permittivity ϵ_r is around 3.3. The edge 2 is located at $x = 0.33$, where ϵ_r is around 1. The blue and red curves show the simulated standing wave effects due to edges 1 and 2, respectively. We can see that while there is no visible standing wave near the edge 2, the mismatched permittivity at edge 1 does produce notable standing waves. In both cases, there is no visible impact to the electric field on the left side of the resonance region.

Fig. R5. Simulated results of the reflection on the K-K medium slab.

The second scenario is to absorb wave incident from outside. In this case, in order to obtain a slab-like K-K profile absorber, the permittivity profile should be cut off on both sides of the resonance region. Particularly, it should be cut on its left side at the place where the relative permittivity is close to 1.

We show in Fig. R5(a) how to obtain such a finite-size K-K profile slab based on the concept of “dispersion engineering” [R3]. By constructing a K-K permittivity profile containing two resonance regions, i.e., $\varepsilon(x)=1-x_{p1}^2/(x^2+i\gamma_1x-x_1^2)-x_{p2}^2/(x^2+i\gamma_2x-x_2^2)$, a unity relative permittivity must exist between locations x_1 and x_2 . If $x_{p1} = 2\lambda_0$, $\gamma_1 = 0.2\lambda_0$, $x_1 = 1\lambda_0$, $x_{p2} = 4\lambda_0$, $\gamma_2 = 0.2\lambda_0$ and $x_2 = 12\lambda_0$, the location with $\varepsilon = 1$ is at $x = 5.45\lambda_0$. If the front edge is cut off at this location, and the back edge can be cut off at a location on the right side of the second resonance region, such as at $x = 13.45\lambda_0$ where $\varepsilon_r = 0.55$, a slab with a thickness of $8\lambda_0$ can be obtained.

Figure R5(b) shows the full-wave simulation to this slab when a point source is placed $2\lambda_0$ away from the left edge. We see that there is no visible standing wave effect due to the front cutting edge.

According to the above discussion, the absorbers implemented with the K-K profile media would have unique mechanisms compared with conventional ones. In the revision, we have these differences. Please refer to our revised manuscript for details.

[R3] Ye, D., Wang, Z., Xu, K., Li, H., Huangfu, J., Wang, Z. & Ran, L. Ultrawideband dispersion control of a metamaterial surface for perfectly-matched-layer-like absorption. Phys. Rev. Lett. 111, 187402 (2013).

Specific remarks:

page 1:

- very recent research -> recent research

- in a good agreement -> in good agreement

We have corrected these in the revision. Thank you

- mathematically speaking eq. 2 has two resonance: $\omega = +$ and $-\omega_0$

Thank you for your reminder. We have clarified this in the revision.

page 2:

- figure 1: Why is the color scale in (b) stating for the imaginary part at -2 when the lowest values is 0, also it indicates negative values for $\text{im}(\varepsilon)$ which is not possible.

Thank you for pointing out this negligence. We have corrected the lowest value to 0.

It would be helpful to plot the cut at $y=0$ for the $|E_z|$ maybe in log scale to indicate how high the transmission is (in dB) an important quantity for an absorber. Especially as in an experiment finite size problems will occur. Also the decay that is due to the point/line source should be indicated ($1/R$) as oppose to no decay for a plane wave.

Fig. R6. Updated figure 1c in this revision.

Thank you for this suggestion. In the revision, we have added subfigures in the bottom of figure 1c to show the absorption in dB scale along the x direction at $y = 0$ (in green lines), as seen in Fig. R6.

Following your suggestion, free-space decay (due to a point source) and the decay due to the K-K medium have been shown for comparison (in black dashed lines in Fig. R6). It can be seen that the decay due to the omnidirectional transmission is negligible compared with that due to the absorptive K-K profile medium.

What are the boundary conditions (at $\pm 10\lambda_0$) for the calculations (pml)?

In our simulations, all simulation regions in the manuscript were truncated by the perfectly matched layers.

page 3:

- Same problem with color scale in figure 2e as in figure 1b

We have corrected this in the revision.

- Parameters of the obtained Lorentz lines should be given

In our work, the obtained space-frequency Lorentzian dispersion described by equation 2 can be considered as $\epsilon(\omega, x) = 3 - (0.57e9)^2 / (\omega^2 + i \times (1.8e7) \times \omega - (2.657e9 - (0.5e7/a) \times x)^2)$, where x and ω range from 0 to $90a$ and 2 to 2.8 GHz, respectively. We have clarified this in the revision.

page 5:

- The structure itself is not rotational invariant but is only invariant on rotations of 90° . Might not be clear to all readers why its rotationally invariant for the field.

We agree that the expression of “rotationally symmetric” may cause potential confusion. As you indicated, the element is only invariant on rotations of 90° . We have clarified this in the revision.

page 7:

- How close can the line source be placed to the resonant part?

Thank you for this good question.

In theory, all waves' incident from the left side of the resonance region can be completely absorbed without any reflection, no matter how close the line source is placed.

page 8:

- Are the bottom and top plate larger than the metamaterial? When yes how many λ s to the border? Are other absorber used or is the whole setup in free space?

Yes, both copper plates are slightly (around one wavelength) larger than the fabricated sample. The whole setup is in free space without using any other absorber.

In the revision, we have clarified this. Please refer to our revised manuscript for details.

- How is direct coupling by evanescent wave taken into account from 1 monopole antenna (source) to the scanning monopole antenna if they are close?

Thank you for this good question.

The periodicity of the meshes in the experimental setup is around $\lambda_0/10$. Therefore, the closest distance between the line source and the probe antenna is also around $\lambda_0/10$. Normally, evanescent couplings are considered to exist inside the reactive near-field range defined by $\lambda_0/6$. Therefore, only the data measured in the eight meshes immediately surrounding the line source could include some evanescent couplings. Compared with the size of the fabricated sample, the introduced error in such a small region can be neglected.

page 9:

- The simulations shown in figure 5c use which boundary conditions (pml or free space to be more adjusted to the experiment).

The same as indicated in figure 5c (Fig. 4c in the original version) in the revision, a free space boundary was used at the cutting edge, and a PML boundary was used to enclose the entire simulation region. Such a boundary conditions setup is all the same to that for experiment.

Reviewer #3

The manuscript demonstrates experimentally at microwaves the recent theoretical finding (from the authors) that absorption over a range of frequencies can happen without reflection in a passive inhomogeneous media provided the permittivity profile of the medium satisfies the spatial Kramers-Kronig (K-K) relations. The topic may be interesting not only from the physics (fundamental) point of view, but also from an engineering (application) perspective. The manuscript is pedagogically written and the work is sound. The measurements support nicely the theoretical frame. It is for these reasons that I am inclined to recommend its publication in Nature Communications.

However, the contribution to the state of the art of this work is simply the experimental demonstration; theory and simulations are not new; I wonder if this is enough for Nature Communications. Next I list in order of importance the points that the authors should address in the revision version to have my full support.

Thank you for your review and your encouraging comments.

Concerning the significance of this work, we proposed the concept of “space-frequency Lorentzian dispersion” that can be used to practically implement a wideband K-K profile medium. Our experimental results also verified the theoretical predictions made in reference 9. As we have replied to Reviewer 1, this K-K profile medium is by far the only inhomogeneous medium that can exhibit omnidirectionally reflectionless absorption without gain.

Compared with existing absorbing materials, a unique characteristic of this medium is that it has a omnidirectionally matched absorbing region. In theory, all energy of a wave entering into in this region from its left side will be absorbed without any reflection, no matter the angle of incidence. As shown in the revised Fig. 1c, a sharp drop of incident intensity up to 120-dB can be observed in this narrow region. To the best of our knowledge, this kind of absorption cannot be implemented by existing absorbers.

In the revision, we have further clarified and emphasized these points. Please refer to our revised manuscript for details.

The authors suggest that this work should have an impact in computational electromagnetics, namely for PML. However, the work does not discuss the common scenario where PML is used: an interface between the PML and a region with fields. As a matter of fact, if the excitation is outside the inhomogeneous medium, all the properties are lost. I think this should be discussed to avoid misleading the readers.

Thank you for this good question.

As we have replied to the Reviewer 2, there are two different scenarios in the usage of the finite-sized K-K profile media.

The first scenario is to absorb waves radiated from inside. In this case, the absorber can be designed following the method used in our experimental setup. As discussed, when the permittivity profile is cut on the right side of the resonance region, for radiations coming from its left side, this region can be considered as an omnidirectionally reflectionless absorbing boundary. Multiple such boundaries can be

used to enclose the space containing radiative sources. For this absorber, according our experimental results, the finite size effect can be negligible.

Fig. R7. Simulated results of the reflection on the K-K medium slab.

The second scenario is to absorb wave incident from outside. In this case, in order to obtain a slab-like K-K profile absorber, the permittivity profile should be cut off on both sides of the resonance region. Particularly, it should be cut on its left side at the place where the relative permittivity is close to 1.

We show in Fig. R7(a) how to obtain such a finite-size K-K profile slab based on the concept of “dispersion engineering” [R3]. By constructing a K-K permittivity profile containing two resonance regions, i.e., $\epsilon(x)=1-x_{p1}^2/(x^2+i\gamma_1x-x_1^2)-x_{p2}^2/(x^2+i\gamma_2x-x_2^2)$, a unity relative permittivity must exist between locations x_1 and x_2 . If $x_{p1} = 2\lambda_0$, $\gamma_1 = 0.2\lambda_0$, $x_1 = 1\lambda_0$, $x_{p2} = 4\lambda_0$, $\gamma_2 = 0.2\lambda_0$ and $x_2 = 12\lambda_0$, the location with $\epsilon = 1$ is at $x = 5.45\lambda_0$. If the front edge is cut off at this location, and the back edge can be cut off at a location on the right side of the second resonance region, such as at $x = 13.45\lambda_0$ where $\epsilon_r = 0.55$, a slab with a thickness of $8\lambda_0$ can be obtained.

Figure R7(b) shows the full-wave simulation to this slab when a point source is placed $2\lambda_0$ away from the left edge. We see that there is no visible standing wave effect due to the front cutting edge.

According to the above discussion, the absorbers implemented with the K-K profile media would have unique mechanisms compared with conventional ones. In the revision, we have discussed these differences. Please refer to our revised manuscript for details.

[R3] Ye, D., Wang, Z., Xu, K., Li, H., Huangfu, J., Wang, Z. & Ran, L. Ultrawideband dispersion control of a metamaterial surface for perfectly-matched-layer-like absorption. Phys. Rev. Lett. 111, 187402 (2013).

Can the authors please quantify the drop in intensity for each frequency? For example, what is the intensity at the edges (Fig. 5) compared to the excitation point? dB drop of?

Thank you for this good suggestion.

Fig. R8. Simulated and measured intensity drop along the x direction at $y = 0$.

In the revision, we have added three subfigures as Fig. 6d (Fig. 5 in the original version), to show the drop of intensity quantitatively. For your convenience, such figures are also shown in Fig. R8.

We can see that in the three cases, the experimental intensity drops measured at the corresponding edge are 40 dB, 41 dB and 39 dB for 2.3, 2.4 and 2.5 GHz, respectively, while all the simulated intensity drops are around 45 dB.

The authors should discuss the potential coupling between consecutive elements that may modify the results shown in Fig. 2d,e. Can the authors unambiguously state that local periodicity condition applies in their designs.

Thank you for this good question.

It has been recognized that the construction of homogeneous effective media relies on the periodic sub-wavelength resonances. The mutual coupling between consecutive elements is not only inevitable, but also used in obtaining macroscopic effective parameters such as permittivity and permeability. The coupled fields between the adjacent elements let the distribution of the total field in between all the elements become flatter. This is how the effective parameters can be macroscopically observed.

In the construction of inhomogeneous effective media such as the K-K profile medium and the invisibility cloaks, the only difference is that the consecutive elements work at slightly changed frequencies discretized from a predefined spatial dispersion. The coupled fields become slightly weaker, but still average the total field between adjacent elements. As a result, when the changes between adjacent elements are small enough, the averaged total field would exhibit a continuous distribution in the macroscopic scale, instead of a constant one in the homogeneous case.

According to the above understanding, the construction of inhomogeneous effective media also relies on the mutual couplings between consecutive elements.

The manuscript does not give enough technical details (material properties, boundary conditions, mesh, etc.) about the simulations to replicate the results precisely. Please, add such details (perhaps as a supplementary material). How many unit cells are stacked in z? The text does not say anything and one may think it is just one, but one can see at least two in Fig. 4d. Is the response of the monopole and probe de-embedded in the results? What is the experimental calibration procedure?

In the revision, we have provided the detailed information about material properties, boundary conditions, mesh and etc. The 2 unit cells stacked in the z direction was also clarified.

In our experiment, an in-system calibration is used to de-embed the response of the monopoles. It was performed by normalizing all the measured data by the amplitude and phase of the electric field measured at one unit cell nearest to the source monopole.

In the revision, we have added these technical details. Please refer to our revised manuscript for details.

The authors claim that only TEM modes are supported. How do they rule out modes like magnetoinductive waves (this is related to the potential mutual coupling between consecutive elements)?

Thank you for this question.

Since the fabricated sample with $\lambda_0/5$ height was sandwiched by two metallic surfaces, for the used frequencies, only TEM mode can exist. This can be clearly seen in the full wave simulation.

In the meantime, since there is no magnetic loop exists in each element or two consecutive elements, the condition of the potential existence of magnetoinductive waves seems not existent. They cannot be observed in full wave simulations either.

The design is for Ez-polarized waves and it will not work for Hz-polarized waves. This should be stated clearly in the text.

Thank you for this suggestion. We have clarified this in the revision.

Can the authors quantify the standing wave ratio (SWR) for each case in Fig. 6?

Thank you for the good suggestion.

We have calculated the SWR for three cases in Fig. 7 (Fig. 6 in the original version). The maximum SWRs for Figs. 7a, 7b and 7c are 1.02, 1.16 and 1.85, respectively. We have added this data in the revision.

Please, add the dimensions units in 4c and 5.

We have added the dimensions in the revised Figs. 5 and 6 (Figs. 4 and 5 in the original version).

Reviewers' comments:

Reviewer #1

- only submitted Remarks to the Editor -

Reviewer #2 (Remarks to the Author):

The questions raised in my first report have been appropriately taken into account. The presentation has been improved a lot making also the advantage of the proposed absorber more pronounced.

From my side the paper fulfils the standards of Nature Communication and I support publication in Nature Communications.

Reviewer #3 (Remarks to the Author):

I am satisfied with the answers given by the authors. Although I am not fully convinced that an experimental validation justifies publication in a high-impact journal like Nature Communications, I do not have a major objection to it.

I want however to drop few extra comments for the authors:

I simply wonder whether a gradient index structure (linear gradient, instead of Lorentzian dispersion) cannot offer the same reflectionless absorption. Perhaps this simpler approach is not so broadband; it would be good to quantify the limitation of this straightforward approach and introduce a discussion.

Figure 2: I think the discussion needs to be elaborated a bit further. It could be as simple as plotting as well $w=1.2w_0$ (the other extreme case of Fig. 1c), E_z in dB as a function of the Gaussian beam propagation direction, and give some quantitative values.

Is the dielectric substrate FR-4? In the text and in the caption of Fig. 3 one can read F4 though.

I am satisfied with the answers given by the authors. Although I am not fully convinced that an experimental validation justifies publication in a high-impact journal like Nature Communications, I do not have a major objection to it.

I want however to drop few extra comments for the authors:

I simply wonder whether a gradient index structure (linear gradient, instead of Lorentzian dispersion) cannot offer the same reflectionless absorption. Perhaps this simpler approach is not so broadband; it would be good to quantify the limitation of this straightforward approach and introduce a discussion.

Thank you for raising this good point.

In microwave engineering, linear gradient index has been widely used. Examples include the impedance matching obtained by continuous impedance transform and the absorbing pyramids used in anechoic chambers. Compared with the case of Lorentzian dispersion, the most significant difference is that the linear gradient index does not have a resonance region. As such, the linear gradient index medium does not have a matched “absorbing boundary” that can exhibit reflectionless absorption over a short distance. This is the reason why a practical continuous impedance transform segment always requires a sufficient length to lower down the slope of the linear gradient index.

Figure R1 | Reflection comparison between K-K and linear profile slabs. (a) and (b) are the Figs. 8(b) and 8(c) of the original manuscript, showing a finite-size K-K profile slab and its reflection characteristic. (c) The linear profile slab with the same thickness and the variation range of the imaginary part of permittivity. (d) The reflection due to this linear profile slab.

Figure R1 shows the comparison between K-K and linear profile slabs, in which (a) and (b) are the Figs. 8(b) and 8(c) of the original manuscript, showing the finite-size K-K profile slab and its reflection characteristic. Figure R1(c) shows the linear profile slab with the same thickness and the same variation range of the imaginary part of permittivity. Note that in Fig. R1(c), the real part has been set to 1, which presents the best case for an absorbing medium with linear profile index. Figure R1(d) shows the reflection due to such a linear profile slab. We can see that a significant reflection can be

observed. Further simulation shows that only when the thickness increases to ~ 90 wavelengths, the reflection can be reduced to a negligible level.

In the revision, we have added this figure and extended a brief discussion. Please refer to our revised manuscript for details.

Figure 2: I think the discussion needs to be elaborated a bit further. It could be as simple as plotting as well $w=1.2w_0$ (the other extreme case of Fig. 1c), E_z in dB as a function of the Gaussian beam propagation direction, and give some quantitative values.

Thank you for this suggestion.

In the revision, we have added a new subfigure as Fig. 2b to elaborate this discussion, where the frequency has been set to $1.2\omega_0$. For your convenience, the new Fig. 2 is posted below.

We see that in all cases, the Gaussian beams are continuously dissipated and finally absorbed in the corresponding resonance regions without any observable reflection, and all the intensity drops in such resonance regions are larger than 100 dB.

Figure R2 | The new Fig. 2 updated in the revision.

Is the dielectric substrate FR-4? In the text and in the caption of Fig. 3 one can read F4 though.

FR-4 was not used in this work. The dielectric substrate used in our work is F4. It has a dielectric constant around 2.55.

REVIEWERS' COMMENTS:

Reviewer #3 (Remarks to the Author):

The manuscript is ready for publication